# Sol–Gel Ceramics for SEIRAS and SERS Substrates

**Jesús Alberto Garibay'Alvarado and Simón Yobanny Reyes-López ***

Departamento de Ciencias Químico Biológicas, Instituto de Ciencias Biomédicas, Universidad Autónoma de Ciudad Juárez, Chih 32315, Mexico; jesus.garibay@uacj.mx
* Correspondence: simon.reyes@uacj.mx

**Abstract:** Surface Enhanced Infrared Absorption Spectroscopy and Surface Enhanced Raman Spectroscopy are analytic techniques that have not been massively adopted since there are issues that still need to be solved with regard to the nature of the signal enhancement substrates used. The sol–gel method for the obtention of ceramics provides an alternative for the production of said substrates. Ceramics are very wear- and heat-resistant, properties that can be used for their regeneration, and through the sol–gel method, ceramics can be produced with high purity as well as can be fashioned in many ways through different techniques, which can be helpful in the pursuit of reproducibility. This paper discusses the different advantages of sol–gel ceramics, their use in the electrospinning technique, and their application in infrared and Raman surface-enhanced spectroscopy.

**Keywords:** sol–gel; ceramics; SEIRAS; SERS; spectroscopy

## 1. Introduction

The process of inorganic polymerization or the sol–gel method is an inexpensive, low-energy consuming, and stable process for obtaining high-purity ceramics, which offer versatility for the production of different kinds of devices, some of which can be used in surface-enhanced spectroscopy such as Surface Enhanced Infrared Absorption Spectroscopy (SEIRAS) and Surface Enhanced Raman Spectroscopy (SERS) [1–3]. The sol–gel process uses reactants such as alkoxides and metal salts involved in a relatively easy hydrolysis reaction [4]. The alkoxides or salts are partially hydrolyzed and then polymerized through condensation, allowing the formation of a tridimensional structure, a gel [5]. Sol–gel-derived ceramics, glasses, and composites offer distinct properties that can be taken advantage of in order to shape or fashion materials with uses ranging from thin films to matrixes, fibers, and monoliths, amongst others [6,7]. A variety of uses such as corrosion protection [8], antibacterial activity [9], bioactive behavior [10], energy storage [11], and catalytic activity [12] can be achieved. Sol–gel ceramic materials can be tailor-made in the nanoscale dimension [13]. Single component or composites [14] allow for exploitation of the properties of the constituents as well as the properties of the ultrastructure of the material [15].

Through various synthesis and surface modification techniques that focus on the production of nanostructures with complex shapes, it has been possible to obtain amplification surfaces with different degrees of success for spectroscopy [16]. The shape of the nanostructures can increase the number of hot spots on the amplification surface, but the function ultimately depends on the reproducibility of the nanostructures and the response, the characteristics of the materials used such as chemical and mechanical resistance, as well as the cost–benefit ratio of the materials used. The use of sol–gel ceramic precursors and the electrospinning technique make the production of fibrous ceramic enhancement substrates for infrared and Raman spectroscopy possible [17]. Such materials have an increased surface area, which provides the capability to enhance spectroscopy signals due to better contact with the analytes [18].

Sol–gel ceramics can also provide the chemical and mechanical resistance much needed for reuse of the substrates [19]. In this review, we discussed the use of the sol–

gel method for the obtention of ceramics such as silica, zirconia, titania, hydroxyapatite, lithium niobate, and alumina and their role in the production of enhancement substrates. Lastly, some advances in this research path have been highlighted.

## 2. The Sol–Gel Method

A sol is a colloid of particles so small that they are suspended usually in a liquid because of the low gravitational and intermolecular forces. If the particles form a tridimensional network, then the sol forms a gel, but sometimes, a precipitate is formed instead [20]. Hench and West described seven steps in the process for obtaining silica monoliths by sol–gel using colloidal powders or alkoxides [21], which can also be applied to other types of materials, in particular, ceramics. The process of inorganic polymerization has been described as stable and inexpensive with low energy requirement as well as versatile since its product can be subsequently used for many purposes [22]. The production of glasses and ceramics benefits greatly from the process because it requires less energy [23], the structure of the material can be controlled at a very small scale, and multiple sol–gel precursors can be used to produce composites [24].

The process starts when the colloidal powder is mixed with a solvent or the alkoxide is hydrolyzed. The low-viscosity liquid is casted into a mold and then left to rest to achieve gelation and to increase its viscosity. At this moment, the gel can be processed to obtain diverse devices. For the objects to maintain its form, the gel must be aged, and in this step, polycondensation occurs, increasing the number of bonds between the particles or the species and the number of pores. Drying is necessary to remove any solvent from the gel; this part of the process must be controlled to avoid rupture of the gel. Chemical stabilization is needed to remove chemical groups capable of undergoing reactions. Finally, to densify the porous gel, it must be heated and, depending on the composition, the temperatures will vary from hundreds to more than a thousand degrees Celsius [21] (Figure 1).

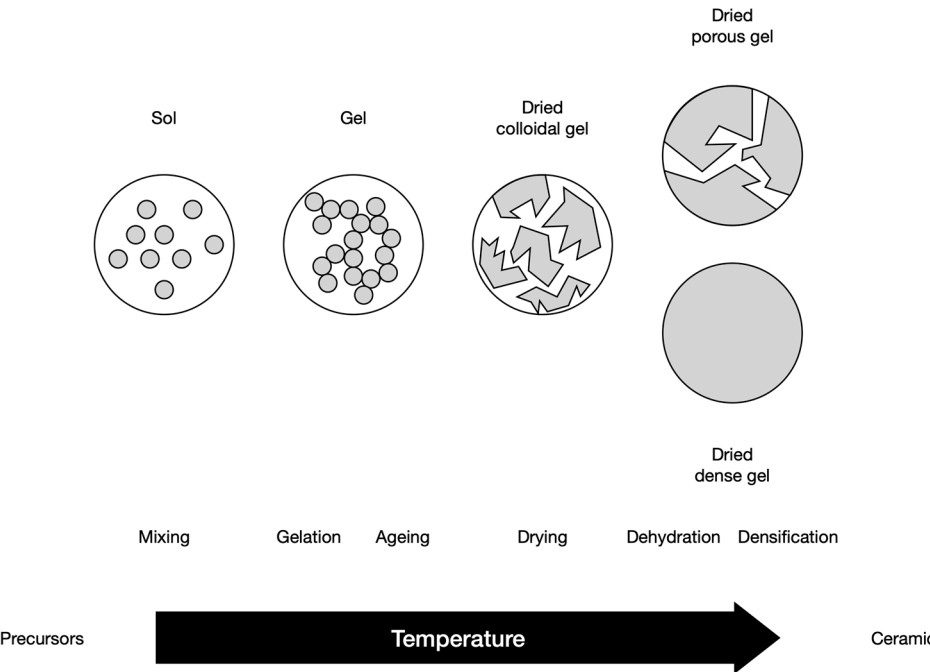

**Figure 1.** Schematic representation of the gel evolution during the sol–gel process.

### 2.1. Hydrolysis and Condensation

Most metal alkoxides are suitable for use in the production of sol–gel ceramics since water can be used to change the molecular structure through hydrolysis [25] because the intention of the process is for the metal atom to be surrounded by hydroxyl ions as a result of severing the alkoxide alkyl chain, and in turn, the same hydroxyl ions will be participants

of a condensation reaction to join the hydrolyzed metal-containing molecules [26]. An acid, a base, or a nucleophilic substance [22] can be used as a catalyst to control the rate of rupture of the alkoxide molecules and to sometimes aid in polycondensation [27]. Obtaining silica through alkoxide hydrolysis is a process very well understood. It was the example used in the legendary article by Hench and West [21] (Figure 2), but over the years, the process has been modified to suit any number of needs such as in the production of bioactive powders [28], functionalized silica for the adsorption of metallic ions for environmental purposes [29], silica–zirconia composite membranes for gas separation [30], and silica–graphene oxide nanocomposites for the improvement of mechanical properties of natural rubber [31], amongst other examples.

$$\equiv\text{Si-OR} + \text{H}_2\text{O} \quad \rightarrow \quad \equiv\text{Si-OH} + \text{ROH} \qquad \text{Hydrolysis}$$
$$\equiv\text{Si-OH} + \text{Si-OR} \quad \rightarrow \quad \equiv\text{Si-O-Si}\equiv + \text{ROH} \qquad \text{Condensation}$$
$$\equiv\text{Si-OH} + \equiv\text{Si-OH} \rightarrow \quad \equiv\text{Si-O-Si}\equiv + \text{H}_2\text{O} \qquad \text{Condensation}$$

**Figure 2.** The process of hydrolysis and condensation using a silicon alkoxide.

Zirconia, titania, hydroxyapatite, and lithium niobate have also been obtained with the sol–gel method via the use of alkoxides (Table 1). Examples such as a zirconia membrane for the separation of hydrogen from a mixture of gases [32], zirconia–silver oxide nanoparticles for antibacterial applications [33], titania–polymer thin film composites for catalytic activity [34], titania thin films doped with zinc and nitrogen with bactericidal and photocatalytic activities [35], hydroxyapatite powder doped with silver nanoparticles with antimicrobial activity [36], bioactive poly-$\varepsilon$-caprolactone with alumina and hydroxyapatite composite fibers [37], lithium niobate thin films with optical properties [38], lithium niobate thin films with piezoelectric properties [39], and copper oxide-lithium niobate composites with photocatalytic activity for environmental purposes prepared through the Pechini method [40]. Examples of alumina prepared using alkoxides include the synthesis of $\gamma$-alumina using aluminum sec-butoxide [41], $\gamma$-alumina microspheres [42], alumina thin films on steel for corrosion protection using aluminum isopropoxide [43], and the use of aluminum triethoxide to improve the properties of a graphene oxide nanocomposite [44]. Yoldas [45,46] made use of aluminum alkoxides hydrolyzed with water and strong acids to help in condensation to produce alumina through the sol–gel method.

**Table 1.** Examples of ceramics obtained by the sol–gel method.

| Ceramic | Precursor | Reference |
|---------|-----------|-----------|
| Silica | Tetramethyl orthosilicate | [47] |
| Silica | Tetraethyl orthosilicate | [48] |
| Zirconia | Zirconium butoxide | [49] |
| Zirconia | Zirconium butoxide | [50] |
| Titania | Titanium isopropoxide | [51] |
| Titania | Titanium isopropoxide | [52] |
| Hydroxyapatite | Triethyl phosphite | [53] |
| Hydroxyapatite | Triethyl phosphite | [54] |
| Lithium niobate | Dihydrate lithium acetate and niobium chloride | [55] |
| Lithium niobate | Lithium carbonate and niobium pentoxide | [56] |
| Alumina | Aluminum nitrate | [57] |
| Alumina | Aluminum nitrate | [58] |

Metal salts can also be used. They are dissolved in an aqueous medium, allowing dissociation of the ions [59]. In the case of elements from groups I and II, where alkoxides are rarely pure, it is possible to obtain a pure ceramic since this dissociation helps in removing the metal ion from the whole salt molecule to produce a pure ceramic oxide [27]. The use of salts of transition or other metals is also possible. When the salt is dissociated, the metal ion is solvated, forming a bond with a water molecule, followed by deprotonation of the cation and the formation of either hydroxide or an oxide [60].

Alumina has been extensively prepared with the sol–gel method using metal salts. Aluminum nitrate tetrahydrate is usually dissolved in water or an alcohol. Thiruchitrambalam et al. [61] synthesized alumina using aluminum metal treated with nitric acid for the obtention of aluminum nitrate and then alumina. Behera et al. [62] used aluminum nitrate and citrate to produce alumina successfully and were able to estimate the optimal citrate/nitrate ratio for nanoparticles to be formed. Alumina prepared from aluminum nitrate has been used for different applications such as a bioactive composite with hydroxyapatite [63], $\gamma$-alumina for the adsorption of lead, cadmium and chromium for environmental applications [64,65], $\alpha$-alumina nanoparticles obtained using an auto combustion method with aluminum nitrate and urea [66], and yttrium–alumina films doped with different concentrations of europium on silica for its luminescence performance [67].

Catalysts used in the sol–gel method have an important role in the condensation part since they promote the completion and lead to fast rates of hydrolysis [20]. After hydrolysis, reactive hydroxy groups are formed, and this process facilitates the condensation or polycondensation reactions [26]. The condensation is also greatly influenced by the pH of the reaction (Figure 3). The use of acids increases the rate of hydrolysis and condensation, allowing the formation of large molecules with an open structure because the hydrolysis occurs at the end groups with little branching [68] whereas the use of bases as catalyzers results in hydrolysis in the middle groups and condensation, giving smaller highly branched molecules, which later may be interlinked in the gelation step [69] (Figure 4). During condensation, small molecules are formed instead of a monomer condensation, highly condensed clusters are formed, and then polymerization occurs between them [70].

**Figure 3.** Different mechanisms of reactions of the hydrolysis catalyzed by an acid and a base.

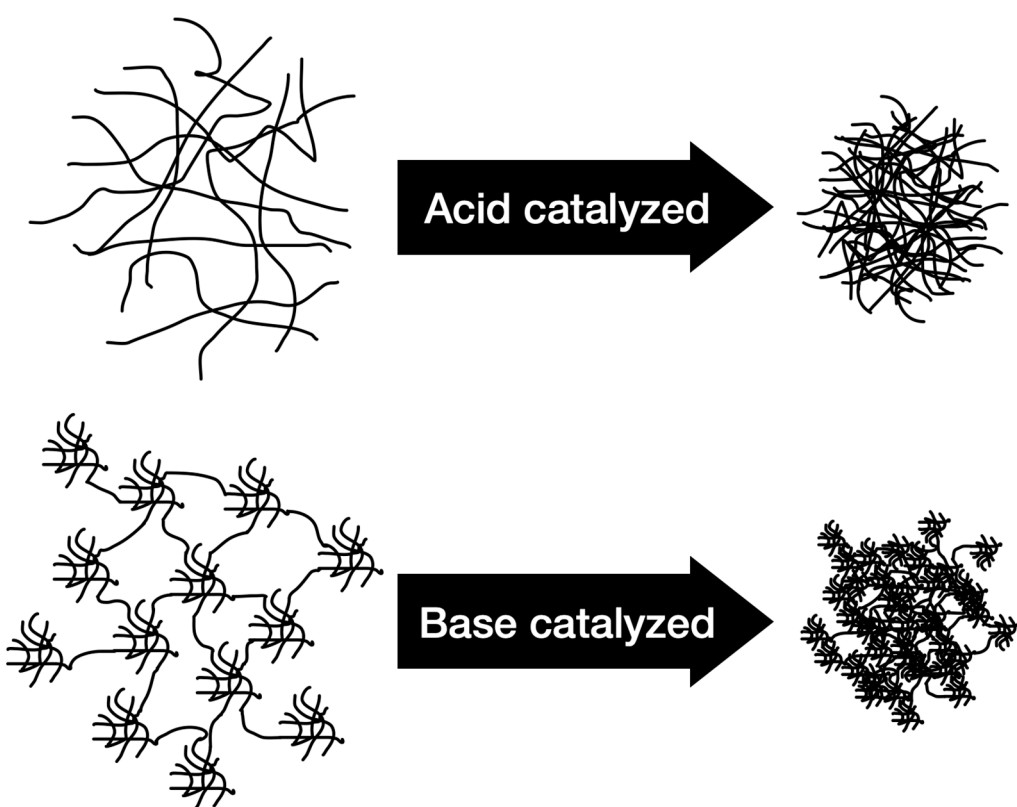

**Figure 4.** Polymerization behavior after hydrolysis catalyzed with an acid and a base.

## 2.2. Gelation

The gelation stage depends on temperature and pH since both parameters have a strong influence on hydrolysis and condensation [71]. When condensation reaches a critical point, gelation occurs, and while the viscosity of the solution is greatly increased, there are still zones in which the sol has not been polymerized yet and clusters of already condensed particles cross-linked together giving an intermediary between solids and liquid [72]. It can be said that a gel is a two-phase substance (solid and liquid) and a whole molecule [73]. Gelation can be greatly affected by the presence of ionic substances, for example, salt in water. Electrolytes can modify the repulsion between particles, and as such, acids used for hydrolysis must to be precisely controlled to obtain a reproducible gel [74]. The beginning of the gel stage is optimal for shaping the ceramic into different devices, as the viscosity is helpful in maintaining shapes [75]. For example, Azarian and Mahmood [76] used zirconia obtained with zirconium n-propoxide mixed with polyaniline to produce conductive films with potential use for electronics; Guo et al. [77] used an organic sponge impregnated with a ceramic slurry and then soaked in a titania sol; Popescu et al. [78] produced spherical bioactive glass composites using tetraethyl othosilicate (TEOS) as sources of silicon, copper oxide, and alginate; and Jmal and Bouaziz [79] created a glass ceramic based on silica and hydroxyapatite using TEOS and a triethyl phosphite/calcium nitrate for potential bioactive activity.

## 2.3. Aging and Drying

Characteristics such as the porosity and structure of the gel continue to change after the gel phase appears. Reactive groups continue condensing, and a shrinkage in the structure begins. This leads to the separation of any liquid, syneresis. The aqueous media is comprised primarily of water and any solvent used in hydrolysis such as ethanol or any other alcohol [21]. Texture and porosity change even before drying, and some of these characteristics can be influenced by the catalyst, as some dissolution and re-condensation can occur [22]. Aging improves the mechanical characteristics of the gel as well, making it

more resistant. Aging has to be carefully controlled through pH and temperature, since breakage can appear [27]. As a process often overlooked, aging allows the gel to continue polycondensation, increasing the amount of interlinking between particles and reducing the porosity via the growth of inter-particle necks, thus increasing the resistance of the gel [80]. Syneresis increases the amount of water in the medium as a result of condensation between discrete particles that possess the hydroxyl groups in the exterior. Naturally, the size of the gel is reduced and becomes denser (Figure 5) [81]. The use of different times, temperatures, and precursor concentrations during aging bestow certain morphological characteristics on gels such as rippling on thin films [80] or different powder nanoparticle shapes [82] suitable for any number of applications.

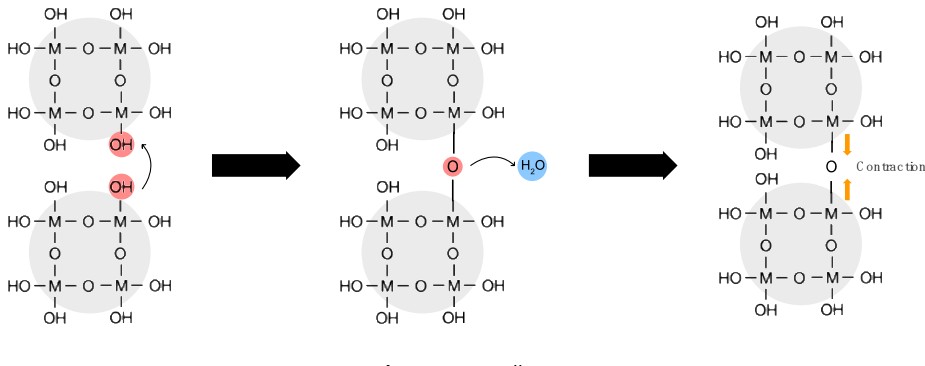

**Figure 5.** Polycondensation between two polymerized particles and the shrinking of the oxo bond.

To remove solvents, drying must be carried out either at room temperature or by using a temperature ramp [83]. This is a critical process that will permit the removal of any solvent, and due to this, any reactions will cease, allowing stabilization of the material [23]. The process of drying must be slow for the gases resulting from the solvent evaporation to escape without cracking the gel; thus, porosity is a great aid in this matter [84]. Double processing, where the gel is allowed to dry and fracture, can be processed to obtain particles that can be resuspended and allowed to gel for a second time [85]. Speaks studied the effect of concentration, aging, and annealing of sol–gel-derived ZnO thin films and found that larger concentrations of the ceramic precursor led to an increase in grain size while reducing the aging time [86]. Ben-Arfa et al. [87] developed a method for fast drying of a sol–gel silica bio-glass, where they observed that the lack of aging results in less polymerization of the material as well as a lesser degree of crystallinity.

### 2.4. Sintering or Densification and Consolidation of the Material

After drying, sintering is a second stage in which molecules again interlink themselves, this time using high temperatures. Three kinds of sintering can occur: dry sintering, where solid particles are joined together; liquid phase sintering, where solid particles are mixed with a small amount of a liquid phase; and reactive sintering, where solid particles of one or more kind(s) react with each other [88]. Densification of the material obeys a process of contraction; the particles diffuse, forming a mass and sometimes reducing the size of the pores but increasing the size of the grains [89]. Different materials require different sintering temperatures in order to achieve proper densification and transformation into a ceramic. Table 2 shows some examples of sintering temperatures for selected ceramics.

**Table 2.** Ceramics obtained by the sol–gel method and the temperatures used in their sintering.

| Ceramic | Sintering Temperature (°C) | Material | Reference |
|---|---|---|---|
| Silica | 600 and 1200 | Silica matrix | [90] |
| Silica | 600 and 1200 | Glass | [91] |
| Zirconia | 700 | Film | [92] |
| Zirconia | 366, 900, and 1150 | Thin film | [93] |
| Titania | 500 | Fibers | [94] |
| Titania | 500 | Films | [95] |
| Hydroxyapatite | 600 | Nanoparticles | [96] |
| Hydroxyapatite | 800 | Porous scaffold | [97] |
| Lithium niobate | 1000 | Film | [98] |
| Lithium niobate | 900 | Substrate | [99] |
| Alumina | 700 | Catalyst | [100] |
| Alumina | 1000 | Nanocrystals | [101] |

## 3. Electrospinning of Sol–Gel Ceramic Precursors

The production of sol–gel ceramics coupled with the electrospinning technique has been helpful in the obtention of composite materials designed for diverse purposes such as tissue engineering [102], energy devices [103], environmental solutions [104], and photocatalysis [105]. The electrospinning technique allows the production of fibers on the nanometric scale [106], providing an increase in surface area [107], which allows contact with analytes, tissues, and any chemical compound alike. The production of ceramic nanofibers with high porosity makes the surface area even greater, with successful results in most cases where applied. Table 3 lists some cases where ceramic nanofibers obtained through the electrospinning technique were produced. Ávila-Martínez et al. [108] produced a material of $ZrO_2$ (Figure 6a) obtained through the sol–gel method using zirconium butoxide as a precursor mixed on polyvinylpyrrolidone (PVP) and electrospun into fibers for the capture of Allura red dye. Average fiber diameters ranged from approximately 112 nm to 360 nm. The fibers had an estimated adsorption of 0.895 g/mg of the dye. Koo et al. [109] created yttria-stabilizaed zirconia nanofibers using metal salts such as zirconium acetate and yttrium nitrate. The fibers had average diameters between 150 and 120 nm. Pescador-Rojas et al. [110] produced a fibrous composite for heat transport applications, with titania obtained by the sol gel-method using titanium (IV) n-butoxide as a precursor and acetic acid as a catalyzer. The $ZrO_2$ sol was mixed with PVP for subsequent electrospinning. The thermal diffusivity of the material was estimated as $1.52 \times 10^{-3} \ cm^2 \ s^{-1}$.

Lopez de Dicastillo et al. [111] produced a $TiO_2$ material of hollow nanofibers or nanotubes using tetrakis (dimethylamide) titanium as a precursor and electrospun with a polymer; then, the polymeric template was removed. The material was tested in concentrations of 150 to 400 µg/mL for antibacterial activity with high or total inhibition of bacteria. Garibay-Alvarado et al. [112] produced a composite of coaxial fibers, with a silica core and a hydroxyapatite sheath. Both materials were obtained by the sol–gel method using TEOS as a precursor for the silica, and triethyl phosphite and calcium nitrate for hydroxyapatite. The fibers had approximate diameters ranging from 510 to 560 nm. Roque-Ruiz et al. [113] produced a fibrous composite consisting of silica and hydroxyapatite obtained through the sol–gel method and then mixed in a single polymeric phase for use in the capture of cadmium and lead from aqueous solution. The precursors used were TEOS and triethyl phosphite/calcium nitrate for silica and hydroxyapatite, respectively. The mean diameter of the fibers was approximately 150 nm. The material achieved an amount adsorbed of $Cd^{2+}$ of 93.3 mg/g and 466.98 mg/g of $Pb^{2+}$ (Figure 6b). Garibay-Alvarado et al. [114] produced a fibrillar material of lithium niobate obtained through the sol–gel method, using

lithium–niobium ethoxide as a precursor and acetic acid as a catalyzer. The resulting sol was mixed at different concentrations with PVP and then electrospun and sintered. Fibers had average diameters ranging from 330 to 760 nm. Wang et al. [115] produced a γ-alumina fibrous material for filtration. Acetic acid, formic acid, and aluminum powder were used in order to obtain aluminum formate. Aluminum formate was then mixed with polyethylene oxide and later sintered at different temperatures. It was observed that, between 700 and 900 °C, the γ phase of alumina appeared and that, at 1000 °C, the δ and α phases appeared. The material was self-standing as γ-alumina, but as δ and α phases, it became brittle. The average diameter varied between 200 to 250 nm as the sintering temperature increased. The filtration efficiency of the fibers was higher than 99.9%.

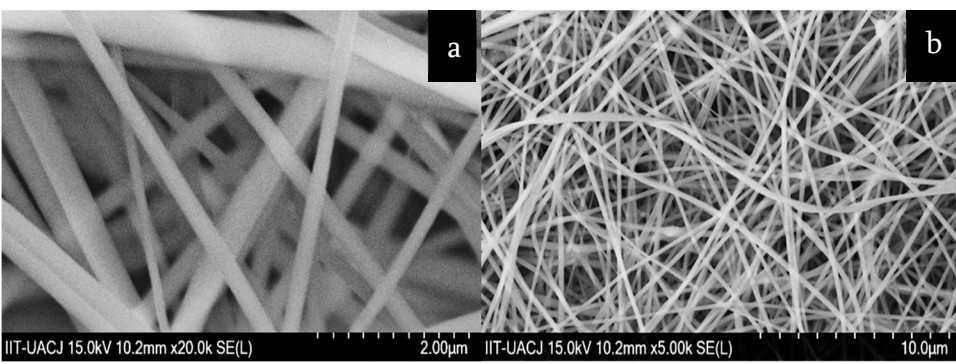

**Figure 6.** $ZrO_2$ fibers for dye capture (**a**) and hydroxyapatite–glass fibers for metal adsorption (**b**).

**Table 3.** Ceramics used in electrospinning for nanofiber obtainment.

| Ceramic | Diameter (nm) | Use | Reference |
|---|---|---|---|
| Silica | ≈500 | Beryllium uptake | [108] |
| Zirconia | ≈360 | Dye sorption | [109] |
| Zirconia | ≤200 | Fuel cells | [110] |
| Titania | ≤80 | Heat transport | [111] |
| Titania | ≈160 | Antimicrobial activity | [112] |
| Hydroxyapatite | ≤330 | Bioactive behavior | [113] |
| Hydroxyapatite | ≈150 | Metal adsorption | [114] |
| Lithium niobate | ≈190 | Piezoelectric behavior | [115] |
| Alumina | ≤250 | Filtration device | [116] |
| Alumina | ≤250 | Catalyst support | [117] |

## 4. Sol-Gel Ceramics for SEIRAS and SERS

Different approaches have been taken to incorporate electrospun ceramic nanofibers on the SEIRAS and SERS technologies in order to resolve some of the issues of the mass adoption of such techniques, for example, the reusability, reproducibility, and stability of the enhancement substrates. Xie et al. [117] developed a material using titanium and fluorine-doped tin oxide sheets as supports. The precursor of choice for the sol–gel titania was titanium butoxide. They were able to obtain two kinds of morphologies: a nanorod array where the surface of the support is covered with rods arranged randomly, mostly pointing up and covered with silver nanoparticles, and a nanopore array, a honeycomb-like structure with more regular features and covered with silver nanoparticles. The materials were capable of detecting concentrations of $5 \times 10^{-12}$ M of 4-mercaptobenzoic acid. The use of titania's photocatalytic activity allowed for the material to be regenerated using UV radiation. Xie and Meng [118] created a material composed of sol–gel titania, a nanotube array covered in silver nanoparticles and decorated with graphene oxide. The addition of

graphene oxide was proposed as a solution for increased adsorption since it can chemically enhance the spectroscopy signals in comparison with a $TiO_2/Ag$ material. The material was successful in enhancing the Raman signals of bisphenol A concentrations as low as $5 \times 10^{-7}$ M.

Roque-Ruiz et al. [119] developed a fibrous material composed of coaxial fibers with an amorphous silica core and a crystalline $TiO_2$ sheath, mostly anatase (Figure 7a). Tetraethyl orthosilicate and titanium tetra-isopropoxide were used as precursors. The composite was decorated with silver nanoparticles that formed an outer cover of dendritic structures. Concentrations of 1 nM pyridine were tested and different peaks in the Raman spectrum were amplified between 3 and 13 orders of magnitude. Singh et al. [120] developed a material consisting of a silica thin film covered by $TiO_2$ and gold nanoparticles. The titania nanoparticles were obtained using the sol–gel method and deposited on top of a silicon substrate, then covered with a thin film of gold, and annealed in order to form the gold nanoparticles. The substrate was tested for enhancement, achieving an enhancement factor of $10^7$ for R6G and $10^8$ for methylene blue. Prakashan et al. [121] produced a ceramic matrix of $SiO_2$–$TiO_2$–$ZrO_2$ with gold–silver nanoparticles specifically for the detection of vitamin A. The matrix was obtained using a non-hydrolytic version of the sol–gel method, with tetraethylorthosilicate, titanium (IV) isopropoxide, and zirconium (IV) propoxide as precursors, and then, the nanoparticles were mixed into the matrix using sonication. Although the substrate was not tested for enhancement of Raman or infrared signals, it was tested for amplification of UV/Vis spectroscopy utilizing surface plasmon resonance sensing, achieving a detection of concentrations as low as 10 µM of vitamin A. Prakashan et al. [122] also tested a similar $SiO_2$–$TiO_2$–$ZrO_2$ ceramic matrix, but instead of Au and Ag nanoparticles, the gold was substituted for copper, in this case, for the detection of mercury. The spectroscopic technique used was also the sensing of SPR. The material had high selectivity for such metals, achieving detection of concentrations as low as 0.01 µM of Hg. The material could possibly be used for enhancement of infrared and Raman spectroscopy.

Hareesh et al. [123] synthesized a $TiO_2$–$ZrO_2$ thin film with embedded silver nanoparticles for the detection of R6G through SERS. The ceramic film was obtained using the sol–gel method with titanium (IV) isopropoxide and zirconium (IV) propoxide as precursors. Two sols were produced and then mixed with silver nitrate and dimehtylformamide in order to form silver nanoparticles. The substrate was able to detect a concentration of R6G as low as $10^{-18}$ M, as referenced by the article, as a single R6G molecule. Ji et al. [124] used zirconium nitrate as the precursor for producing $ZrO_2$ nanoparticles through the sol–gel method. Sintering temperatures ranging from 450 to 650 °C were used, and two crystalline phases were obtained, tetragonal and monoclinic, with the latter in the highest proportion. The diameter of the particles ranged from 8.1 to 17.6 nm, or an average of 10.5 nm, and such sizes increased with the increase in sintering temperature. The highest enhancement factor of $4.32 \times 10^3$ was achieved through the adsorption of 4-MBA on the surface of the particles and calcination at 500 °C. Hu et al. [125] produced through templating a nanoarray consisting of silver nanopillars in a flower-like disposition covered with anodized aluminum oxide for corrosion protection of the silver. In order to fashion such a material, sol–gel silica was used in one step of a multi-step process as corrosion protection for the aluminum metal sheet used as a substrate. A Raman enhancement test was carried out on the material, detecting R6G concentrations from $10^{-10}$ M to $10^{-7}$ M. Li et al. [126] produced a composite of $\alpha$-$Fe_2O_3$–$SiO_2$–Ag for the detection of pesticides. The iron and silver nanoparticles were obtained using iron (III) hexachloride and silver nitrate, respectively. The silica was obtained through the sol–gel method using TEOS as a precursor and ammonia as a catalyzer. The iron nanoparticles had a cube-like morphology and were covered by a layer of silica; then, the silver nanoparticles coated the exterior of the Fe–$SiO_2$ nanoparticles. The final composite was tested for signal enhancement in analyzing different concentrations of *p*-aminothiophenol ranging from $1 \times 10^{-4}$ M to $1 \times 10^{-4}$ M, and concentrations of thiram ranging from $1 \times 10^{-3}$ M to $1 \times 10^{-7}$ M. The composite achieved a detection of thiram in concentrations lower than 7 ppm, which is lower than

the limit set by the United States Environmental Protection Agency as of 2016 for such a fungicide.

Shi et al. [127] developed a $ZrO_2$–Ag–$SiO_2$ composite for SERS applications. Silica nanoparticles were prepared using TEOS as a precursor and ammonia as a catalyzer. Silver nanoparticles were deposited on top of the $SiO_2$ as an intermediate layer, and then, the $SiO_2$–Ag composite was covered in a $ZrO_2$ layer obtained through the sol–gel method and using zirconium (IV) propoxide as a precursor. The estimated diameter of the silica nanoparticles was 340 nm and that of the silver nanoparticles was 30 nm. The composite had a SERS activity capable of detecting concentrations of 4-ATP and R6G in the ranges of $10^{-9}$ M and $10^{-8}$ M, respectively. Soto-Nieto et al. [128] produced a fibrous composite of silica, hydroxyapatite, and silver for the enhancement of spectroscopic signals in SEIRAS and SERS. Silica and hydroxyapatite were obtained using the sol–gel method with TEOS and triethyl phosphite–calcium nitrate as precursors, respectively. Fibrous mats were fabricated through electrospinning by mixing the sols with PVP and by sintering at temperatures from 200 to 1150 °C. The fibers had an average diameter of 304 nm (Figure 7b). The fibrous mats were doped with silver nanoparticles using electrodeposition and $AgNO_3$ as the precursor. The SEIRAS enhancement factor was estimated as $2.01 \times 10^6$, and the SERS enhancement factor was estimated as $3.46 \times 10^8$.

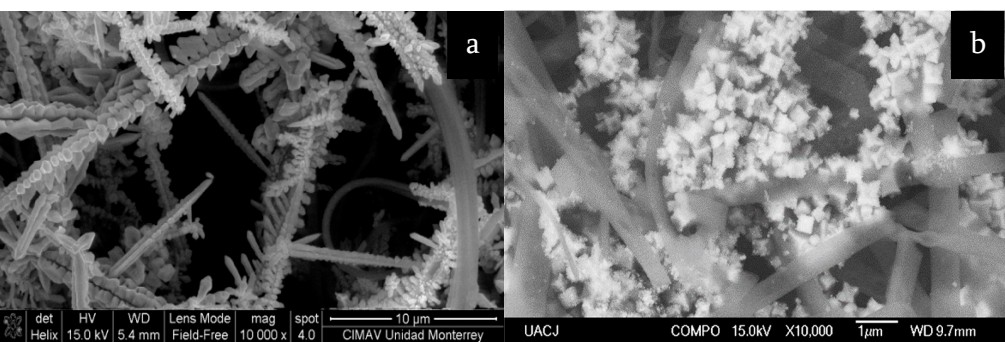

**Figure 7.** $SiO_2$–$TiO_2$–Ag fibrous composite with SERS activity (**a**) and hydroxyapatite–$SiO_2$–Ag fibers with SERS and SEIRAS activity (**b**).

## 5. Conclusions

The use of electrospun ceramic fibers as enhancement substrates is related to the capacity of the ceramic materials to withstand wear, the capacity of some ceramic materials to have a catalytic behavior, and the capability to withstand high temperatures, which can be helpful in the regeneration of such substrates. Porous ceramics can improve contact and adsorb analytes for better interaction. The literature mentioned in this review provides an outlook on the different approaches that have been taken in the use of the sol–gel method to obtain ceramics, how they can be taken advantage of, and some of the initiatives that involve this technique in order to produce devices capable of being used for the amplification of infrared and Raman spectroscopy signals. While there is still much to be explored, the use of the sol–gel method and the electrospinning technique in conjunction are indeed very promising with regard to the production of such devices.

**Funding:** This research received no external funding.

**Acknowledgments:** Thanks to PRODEP, Universidad Autónoma de Ciudad Juárez, and CONACYT.

**Conflicts of Interest:** The authors declare that there are no conflict of interest regarding the publication of this paper.

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
