# Peer review of "Sol–Gel Ceramics for SEIRAS and SERS Substrates"

_crystals, doi:10.3390/cryst11040439_

Round 1
Reviewer 1 Report
The authors use too many references to the work of other researchers, as if they are hedging their every step. I wish you more confidence in your conclusions.
Author Response
The manuscript has been rewritten focusing the detail of the surfaces derived from sol-gel and electrospun to give an amplification effect in SERS and SEIRAS spectroscopies.
Reviewer 2 Report
I have read over the review by Alvarado and Reyes-López entitled, “Sol-gel ceramics for SEIRAS substrates”. I want to first thank the authors for reviewing a subject that is of importance to several scientific communities. Reviewing a subject always requires quite a bit of work and is a service to those communities of scientists and technologists.
However, I do have some concerns about the review that I would like the authors to think about.
First and foremost, I do not understand why a review of glassy ceramics (which are for the most part are completely amorphous) would be submitted to this particular journal, whose focus is on crystalline material research. There seems to be a disconnect here that I will let the editor make the final decision on.
Second, while the title of the review defines the focus area as sol-gel materials for SEIRAS (which is not defined anywhere in the article [?Surface Enhanced IR ?Absorption? Spectroscopy?]). However, upon looking over the organization of the article one cannot help but notice that almost the real first mention of SEIRAS in the article comes in section 4 near the end of page 7 of this 8 page review. Furthermore, section 4 is only one short paragraph long to go with one table (Table 4). This paragraph and table appear to be the entirety of the information presented in the review that is specifically focused on the title subject of the review. I don’t understand this.
The main part of the manuscript (6+ pages of 8, not counting references) is a rather brief overview of the basics of sol-gel reactions and technologies that have been reviewed countless times in the last thirty years. The discontinuity between the defined focus area of the manuscript and what is actually reviewed is a major concern.
My best advice to the authors is to completely rewrite the review and focus in much more detail on the needs of SERS and SEIRAS based spectroscopies and how sol-gel-derived surfaces answer these needs.

Author Response
The manuscript has been rewritten focusing the detail of the surfaces derived from sol-gel and electrospun to give an amplification effect in SERS and SEIRAS spectroscopies.
The manuscript has been rewritten focusing the detail of the sol-gel and electrospun derived surfaces to give an amplification effect in SERS and SEIRAS spectroscopies.
New information is put into the manuscript to supplement paragraphs and tables developed, also the revision of the contribution of expuest sections was made.
The conclusions were reviewed.
Reviewer 3 Report
The review paper is about "Sol-gel ceramics for SEIRAS substrates".
It contains general sol-gel synthesis and sol-gel theory information.
Section 4. Sol-gel ceramics for SEIRAS and SERS is very short, without a critical review of the state of the art literature.
It is advised to rewrite the paper and strengthen section 4, which is proposed as the main topic of this review paper.
Author Response
The manuscript has been rewritten focusing the detail of the sol-gel and electrospun derived surfaces to give an amplification effect in SERS and SEIRAS spectroscopies.
New information is put into the manuscript to supplement paragraphs and tables developed, also the revision of the contribution of expuest sections was made.
The conclusions were reviewed.
Reviewer 4 Report
Having examined your manuscript entitled “Sol-gel ceramics for SEIRAS substrates” I note that you have made an interesting review of sol-gel method applied to ceramics. However, you have not discussed in detail the how the materials are useful for surface enhanced spectroscopy such as SEIRAS and SERS. Therefore, it requires a major revision. Please check carefully.
The author explained vaguely the potential of sol-gel materials for surface enhanced spectroscopy such as SEIRAS and SERS.
The review focus on sol-gel synthesis but only section 4 is dedicated to sol-gel ceramics for SEIRAS and SERS. Few lines are dedicated to SEIRAS and SERS.
It seems that the article discuss the basics of sol gel synthesis in detail but it is not focus on the development on sol-gel material for SEIRAS and SERS.
Please define the requirements of Sol-gel ceramics for SEIRAS substrates and the materials developed for this specific application in detail,
The authors discuss vaguely the applicability of this kind of compounds, and the requirements, stability needed.
Therefore your article will not be considered, if the author does not include and add the changes proposed
Author Response

(The authors gave the same response as above.)

Round 2
Reviewer 2 Report
The authors have improved their review of sol-gel approaches to SEIRAS and SERS substrates. There is more direct description of recent results and advances in the specific area of the review that are now included.
Reviewer 3 Report
Paper looks much better now.
Reviewer 4 Report
The authors addressed the comments proposed.